# One-bit Supervision for Image Classification

**Hengtong Hu[1,2], Lingxi Xie[3], Zewei Du[3], Richang Hong[1,2]\*, Qi Tian[3]**

[1]Key Laboratory of Knowledge Engineering with Big Data, Hefei University of Technology,
[2]School of Computer Science and Information Engineering, Hefei University of Technology,
[3]Huawei Inc.

huhengtong.hfut@gmail.com, 198808xc@gmail.com, duzewei@huawei.com,
hongrc@hfut.edu.cn, tian.qi1@huawei.com

## Abstract

This paper presents one-bit supervision, a novel setting of learning from incomplete annotations, in the scenario of image classification. Instead of training a model upon the accurate label of each sample, our setting requires the model to query with a predicted label of each sample and learn from the answer whether the guess is correct. This provides one bit (yes or no) of information, and more importantly, annotating each sample becomes much easier than finding the accurate label from many candidate classes. There are two keys to training a model upon one-bit supervision: improving the guess accuracy and making use of incorrect guesses. For these purposes, we propose a multi-stage training paradigm which incorporates negative label suppression into an off-the-shelf semi-supervised learning algorithm. In three popular image classification benchmarks, our approach claims higher efficiency in utilizing the limited amount of annotations.

## 1 Introduction

In the deep learning era [18], training deep neural networks is a standard methodology for solving computer vision problems. Yet, it is a major burden to collect annotations for training data. In particular, when the dataset contains a large number of object categories (*e.g.*, ImageNet [3]), it is even difficult for human to memorize all categories [30, 41]. In these scenarios, the annotation job would be much easier if the worker is asked whether an image belongs to a specified class, rather than being asked to find the accurate class label from a large amount of candidates.

This paper investigates this setting, which we refer to as **one-bit supervision** because the labeler provides one bit of information by answering the yes-or-no question. In comparison, each accurate label provides $\log_2 C$ bits of information where $C$ is the number of classes, though we point out that the actual cost of accurate annotation is often much higher than $\log_2 C \times$ that of one-bit annotation. One-bit supervision is a new challenge of learning from incomplete annotation. We expect the learning efficiency, in terms of accuracy under the same amount of supervision bits, to be superior to that of semi-supervised learning. For example, in a dataset with $100$ classes, we can choose to accurately annotate 10K samples which give $10\text{K} \times \log_2 100 = 66.4\text{K}$ bits of information, or accurately annotate 5K samples which give 33.2K bits of information, and leave the remainder to answering 33.2K yes-or-no questions raised by the neural network. To verify its superiority, we asked three labelers to annotate $100$ images ($50$ correctly labeled and $50$ wrongly labeled) from ImageNet in one-bit setting. The average annotation time is $2.72$ seconds per image (with a precision of $92.3\%$). According to [30], the average time for a full-bit annotation is around 1 minute, much higher than $10 \times$ of the one-bit cost. This validates our motivation in a many-class dataset.

---

We notice that one-bit supervision has higher uncertainty compared to the conventional setting, because the supervision mostly comes from guessing the label of each sample. If a guess is correct, we obtain the accurate label of the sample, otherwise only one class is eliminated from the possibilities. To learn from this setting efficiently, two keys should be ensured: (i) trying to improve the accuracy of each guess so as to obtain more positive labels, and (ii) making full use of the failure guesses so that the negative labels, though weak, are not wasted. This motivates us to propose a multi-stage training framework. It starts with a small number of accurate labels and makes use of off-the-shelf semi-supervised learning approaches to obtain a reasonable initial model. In each of the following stages, we allow the model to use up part of the supervision quota by querying, with its prediction, on a random subset of the unlabeled images. The correct guesses are added to the set of fully-supervised samples, and the wrong guesses compose of a set of negative labels, and we learn from it by forcing the semi-supervised algorithm to predict a very low probability on the eliminated class. After each stage, the model becomes stronger than the previous one and thus is expected to achieve higher guess accuracy in the next stage. Hence, the information obtained by one-bit supervision is significantly enriched.

We evaluate our setting and approach on three image classification benchmarks, namely, CIFAR100, Mini-ImageNet and ImageNet. We choose the mean-teachers model [33] as a semi-supervised learning baseline as well as the method used for the first training stage, and compare the accuracy under different numbers of accurate labels. Results demonstrate the superiority of one-bit supervision, and, with diagnostic experiments, we verify that the benefits come from a more efficient way of utilizing the information of incomplete supervision.

The remaining part of this paper is organized as follows. Section 2 illustrates the one-bit supervision setting and our solution, and Section 3 shows experiments on image classification benchmarks. Based on these results, we discuss the relationship to prior work and future research directions in Section 4, and finally conclude our work in 5.

## 2 One-bit Supervision

### 2.1 Problem Statement

Conventional semi-supervised learning starts with a dataset of $\mathcal{D} = \{\mathbf{x}_n\}_{n=1}^{N}$, where $N$ is the total number of training samples, $\mathbf{x}_n$ is the image data of the $n$-th sample. Let $y_n^\star$ be the ground-truth class label[2] of $\mathbf{x}_n$, but in our setting, $y_n^\star$ is often unknown to the training algorithm. Specifically, there is a small fraction containing $L$ samples for which $y_n^\star$ is provided, and $L$ is often much smaller than $N$, $e.g.$, as in Section 3.1, researchers often use $20\%$ of labels on the CIFAR100 and Mini-ImageNet datasets, and only $10\%$ of labels on the ImageNet dataset. That being said, $\mathcal{D}$ is partitioned into two subsets, $\mathcal{D}^\mathrm{S}$ and $\mathcal{D}^\mathrm{U}$, where the superscripts stand for 'supervised' and 'unsupervised', respectively.

The key insight of our research is that, when the number of classes is large, it becomes very challenging to assign an accurate label for each image. According to early user studies on ImageNet [30, 41], it is even difficult for a testee to memorize all the categories. This largely increases the burden of data annotation. In comparison, the cost will become much smaller if we ask the labeler *'Does the image belong to a specific class?'* rather than *'What is the accurate class of the image?'*

To verify our motivation that the new setting indeed improves the efficiency of annotation, we invite three labelers who are moderately familiar with the ImageNet-1K dataset [30]. We run a pre-trained ResNet-50 model on the test set and randomly sample 100 images, 50 correctly labeled and 50 wrongly labeled, for the three labelers to judge if the network prediction is correct. The above configuration maximally approximates the scenario that a labeler can encounter in a real-world annotation process, meanwhile we avoid the labeler to bias towards either positive or negative samples by using a half-half data mix. The three labelers report an average precision of $92.3\%$ and the average annotation time for each image is 2.72 seconds. The accuracy is acceptable provided that an experienced labeler can achieve a top-5 accuracy of $\sim 95\%$, yet a full annotation takes around 1 minute for each image [30], higher than $10\times$ of our cost ($\log_2 1000 \approx 10$). Hence, we validate the benefit in learning from a large-scale dataset.

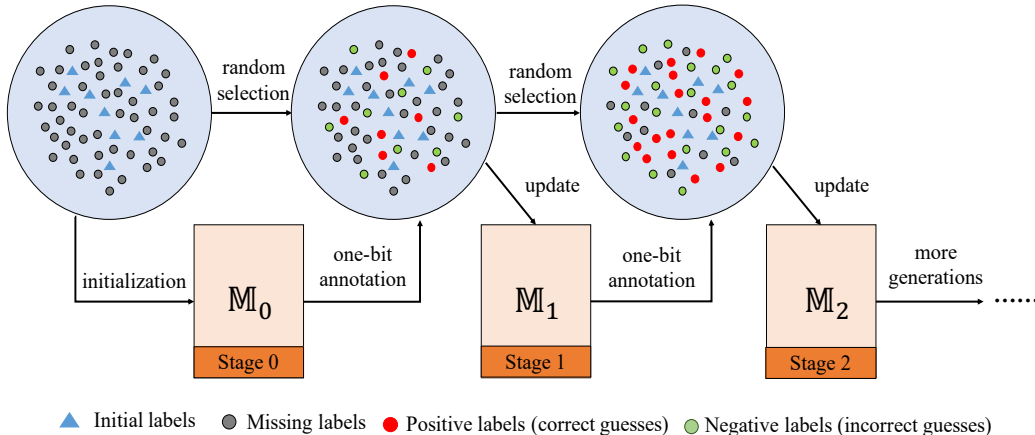

▲ Initial labels  ● Missing labels  ● Positive labels (correct guesses)  ● Negative labels (incorrect guesses)

Figure 1: The training procedure with one-bit supervision (best viewed in color). At the beginning, only a small set of training samples (blue triangles) are provided with ground-truth labels and the remaining part (black circles) remains unlabeled. We initialize the model using an off-the-shelf semi-supervised learning algorithm. In each of the following iterations (we show two iterations but there can be more), part of unlabeled data are sent into the current model for prediction, and the labeler is asked to judge if the prediction is correct. Some of these samples obtain positive labels (red circles) while some obtain negative labels (green circles). This process continues until the quota of supervision is used up as scheduled.

Motivated by the above, we formulate the problem into a new setting that incorporates semi-supervised learning and weakly-supervised learning. The dataset is partitioned into three parts, namely, $\mathcal{D} = \mathcal{D}^{\mathrm{S}} \cup \mathcal{D}^{\mathrm{O}} \cup \mathcal{D}^{\mathrm{U}}$, where $\mathcal{D}^{\mathrm{O}}$ is a fixed set[3] for **one-bit supervision**. For each sample in $\mathcal{D}^{\mathrm{O}}$, the labeler is provided with the image and a predicted label, and the task is to distinguish if the image belongs to the specific label. Note that the guess is **only allowed once** – if the guess is correct, the image is assigned with a positive (true) label, $y_n^\star$, otherwise, it is assigned with a negative label, denoted as $y_n^-$ and no further supervision of this image can be obtained.

From the perspective of information theory, the labeler provides 1 bit of supervision to the system by answering the yes-or-no question, yet to obtain the accurate label, the average bits of required supervision is $\log_2 C$. Therefore, the burden of annotating a single image is alleviated, therefore, with the same cost, one can obtain much more one-bit annotations than full-bit annotations. We use the CIFAR100 dataset as an example. A common semi-supervised setting annotates 10K out of 50K training images which requires $10\mathrm{K} \times \log_2 100 = 66.4\mathrm{K}$ bits of supervision. Alternately, we can annotate 5K images in full-bit and as many as 33.2K images in one-bit, resulting in the same amount of supervision, but with higher learning efficiency[4].

## 2.2 A Multi-Stage Training Paradigm

There are two important factors to one-bit supervision, namely, (i) making high-quality guesses in $\mathcal{D}^O$; (ii) making use of the negative (wrongly predicted) labels. We elaborate our solution for the first factor in this subsection, and leave the second to the next subsection.

Intuitively, the accuracy of a model is improved when it sees more (fully or weakly) labeled training samples. Considering that each image in $\mathcal{D}^O$ can be guessed only once, the straightforward strategy is to partition the training procedure into several **stages**, each of which makes prediction on a part of $\mathcal{D}^O$ and then enhances the model based on the results. This offers the generalized training algorithm as illustrated in Figure 1. The initial model, $\mathbb{M}_0$, is trained with a semi-supervised learning process in which $\mathcal{D}^S$ is the labeled training set and $\mathcal{D}^O \cup \mathcal{D}^U$ is the unlabeled reference set. We use Mean-Teacher [33], an off-the-shelf semi-supervised learning algorithm to utilize the knowledge in the reference set. This provides us with a reasonable model to make prediction on $\mathcal{D}^O$.

The remaining part of training is a scheduled procedure which is composed of $T$ iterations. Let $\mathcal{D}_{t-1}^R$ denote the set of samples without accurate labels before the $t$-th iteration, and $\mathcal{D}_0^R \equiv \mathcal{D}_0^U$. We also maintain two sets, $\mathcal{D}^{O+}$ and $\mathcal{D}^{O-}$, for the correctly and wrongly guessed samples. Both of these sets are initialized as $\varnothing$. In the $t$-th iteration, we first randomly sample $\mathcal{D}_t^O$, a subset of $\mathcal{D}_{t-1}^R$, and use the previous model, $\mathbb{M}_{t-1}$, for receiving one-bit supervision. The strategy of sampling $\mathcal{D}_t^O$ will be further discussed in Section 3.3. Then, we make use of $\mathbb{M}_{t-1}$ to predict the labels of images in $\mathcal{D}_t^O$ and check the ground-truth. The correctly predicted samples are added to $\mathcal{D}^{O+}$ and the others added to $\mathcal{D}^{O-}$. Hence, the entire training set is split into three parts: $\mathcal{D}^S \cup \mathcal{D}^{O+}$ (has full labels), $\mathcal{D}^{O-}$ (has negative labels) and $\mathcal{D}_t^U$ (has no labels), and finally, $\mathcal{D}_t^R = \mathcal{D}_t^{O-} \cup \mathcal{D}_t^U$. To continue iteration with a stronger model, we update $\mathbb{M}_{t-1}$ into $\mathbb{M}_t$ with the currently available supervision. The fully-labeled and unlabeled parts contribute as in the semi-supervised baseline, and we concentrate on making use of the negative labels in $\mathcal{D}^{O-}$, which we will elaborate our solution in the following part.

## 2.3 Negative Label Suppression

To make use of the negative labels, we recall our baseline, the Mean-Teacher algorithm [33], that maintains two models, a teacher and a student, and trains them by assuming that they produce similar outputs – note that this does not require labels. Mathematically, given a training image, $\mathbf{x} \in \mathcal{D}$, we first compute the cross-entropy loss term if $\mathbf{x} \in \mathcal{D}^S \cup \mathcal{D}^{O+}$. In addition, no matter whether $\mathbf{x}$ has an accurate label, we compute the difference between the predictions of the teacher and student models as an extra loss term. Let $\mathbf{f}(\mathbf{x}; \boldsymbol{\theta})$ be the mathematical function of the student model where $\boldsymbol{\theta}$ denotes the learnable parameters. Correspondingly, the teacher model is denoted by $\mathbf{f}(\mathbf{x}; \boldsymbol{\theta}')$ where $\boldsymbol{\theta}'$ is the moving average of $\boldsymbol{\theta}$. The loss function is written as:

$$\mathcal{L}(\boldsymbol{\theta}) = \mathbb{E}_{\mathbf{x} \in \mathcal{D}^S \cup \mathcal{D}^{O+}} \ell(\mathbf{y}_n^\star, \mathbf{f}(\mathbf{x}; \boldsymbol{\theta})) + \lambda \cdot \mathbb{E}_{\mathbf{x} \in \mathcal{D}} \left| \mathbf{f}(\mathbf{x}; \boldsymbol{\theta}') - \mathbf{f}(\mathbf{x}; \boldsymbol{\theta}) \right|^2, \tag{1}$$

where $\lambda$ is the balancing coefficient and $\ell(\cdot, \cdot)$ is the cross-entropy loss. For simplicity, we have omitted the explicit notation for the individual noise added to the teacher and student model. That is being said, the model's output on a fully-supervised training sample is constrained by both the cross-entropy and prediction consistency terms (the idea is borrowed from knowledge distillation [11, 36]). However, for a training sample with a negative label, the first term is unavailable, so we inject the negative label into the second term by modifying $\mathbf{f}(\mathbf{x}; \boldsymbol{\theta}')$ accordingly, so that the score of the negative class is suppressed to zero[5]. We name this method **negative label suppression** (NLS). While there may exist other ways to utilize the negative labels, we believe that NLS is a straightforward and effective one that takes advantages of both the teacher model and newly added negative labels. As we shall see in experiments (Section 3.2), while being naive and easy to implement, negative label suppression brings significant accuracy gain to the one-bit supervision procedure.

From a generalized perspective, NLS is a practical solution to integrate negative (weak) labels into the framework of teacher-student optimization. It is complementary to the conventional methods that used cross-entropy to absorb the information from positive (strong) labels, but instead impact the consistency loss. This is a new finding under the setting of one-bit supervision, and intuitively, as more negative labels are available (*i.e.*, the situation is closer to full supervision), NLS may conflict with the design of teacher-student optimization and weakened cross-entropy may take the lead again.

# 3 Experiments

## 3.1 Datasets and Implementation Details

We conduct experiments on three popular image classification benchmarks, namely, CIFAR100, Mini-Imagenet, and Imagenet. CIFAR100 [16] contains 50K training images and 10K testing images, all of which are $32 \times 32$ RGB images and uniformly distributed over 100 classes. For Mini-ImageNet in which the image resolution is $84 \times 84$, we use the training/testing split created in [28] which consists of 100 classes, 500 training images, and 100 testing images per class. For ImageNet [3], we use the commonly used competition subset [30] which contains 1K classes, 1.3M training images, and 50K testing images, and each class has roughly the same number of samples. All images are of high resolution and pre-processed into $224 \times 224$ as network inputs.

We follow Mean-Teacher [33], a previous semi-supervised learning approach, to build our baseline. It assumes that a small subset, $\mathcal{D}^{S\prime}$, has been labeled. $\left|\mathcal{D}^{S\prime}\right|$ is 20% of the training set for CIFAR100 and Mini-ImageNet, and 10% for ImageNet. We reschedule the assignment by allowing part of the annotation to be one-bit, resulting in two subsets, $\left|\mathcal{D}^{S}\right|$ and $\left|\mathcal{D}^{O}\right|$, satisfying $\left|\mathcal{D}^{S\prime}\right| \approx \left|\mathcal{D}^{S}\right| + \left|\mathcal{D}^{O}\right| / \log_2 C$. The detailed configuration for the three datasets are shown in Table 1. Following [33], we use a 26-layer deep residual network [10] with Shake-Shake regularization [7] for CIFAR100, and a 50-layer residual network for Mini-ImageNet and ImageNet. The number of training epochs is 180 for CIFAR100 and Mini-ImageNet, and 60 for ImageNet. The consistency loss, as in [33], is computed using the mean square error in each stage for all three datasets. The consistency parameter is 1,000 for CIFAR100, and 100 for Mini-ImageNet and ImageNet. Other hyper-parameters simply follow the original implementation, except that the batch size is adjusted to fit our hardware (*e.g.*, eight NVIDIA Tesla-V100 GPUs for ImageNet experiments).

## 3.2 Comparison to Full-bit Semi-supervised Supervision

We first compare our approach to Mean-Teacher [33], a semi-supervised learning baseline in which all annotations are full-bit. Results are summarized in Table 2. One can observe that, the one-bit supervision baseline (with two stages, without utilizing the knowledge in negative labels) produces inferior performance to full-bit supervision. Note that on CIFAR100, our approach actually obtains more accurate labels via one-bit supervision (the number of accurate labels is $3K + 25.3K$, while the baseline setting only has 10K), but the correct guesses are prone to the easy samples that do not contribute much to model training. The deficit becomes more significant in Mini-ImageNet on which the baseline accuracy is lower, and in ImageNet on which the number of incorrectly guessed images is larger. The reason is straightforward, namely, these samples contribute little new knowledge to the learning process because (i) the model has already learned how to classify these samples; and (ii) these samples are relatively easy compared to the incorrectly predicted ones. Therefore, making use of the negative labels is crucial for one-bit supervision.

We next investigate negative label suppression, the basic method to extract knowledge from incorrect guesses. Results are listed in Table 2. One can observe significant improvement brought by simply suppressing the score of the incorrect class for each element in $\mathcal{D}^{O-}$. Compared to the two-stage baseline, this brings 4.37%, 5.86%, and 4.76% accuracy gains on CIFAR100, Mini-ImageNet, and ImageNet, respectively. That being said, the negative labels, though only filtering out one out of 100 or 1,000 classes, can help substantially in the scenario of semi-supervised learning, and the key contribution is to avoid the teacher and student models from arriving in a wrong consensus.

Table 1: The data split for semi-supervised and one-bit-supervised learning, where the total numbers of supervision bits are comparable. We will investigate other data splits in Section 3.3.

| Dataset | $C$ | $\log_2 C$ | $|\mathcal{D}|$ | semi-supervised | | one-bit-supervised | | |
| --- | --- | --- | --- | --- | --- | --- | --- | --- |
| | | | | $\left|\mathcal{D}^{S}\right|$ | # of bits | $\left|\mathcal{D}^{S}\right|$ | $\left|\mathcal{D}^{O}\right|$ | # of bits |
| CIFAR100 | 100 | 6.6439 | 50K | 10K | 66.4K | 3K | 47K | 66.9K |
| Mini-ImageNet | 100 | 6.6439 | 50K | 10K | 66.4K | 3K | 47K | 66.9K |
| ImageNet | 1,000 | 9.9658 | 1281K | 128K | 1276K | 30K | 977K | 1276K |

In summary, with two-stage training and negative label suppression, our approach achieves favorable performance in one-bit supervision. In particular, under the same bits of supervision, the accuracy gain over the baseline, Mean-Teacher, is $4.00\%$, $4.48\%$, and $2.24\%$ on CIFAR100, Mini-ImageNet, and ImageNet, respectively. Provided that the current question (querying whether the image belongs to one specific class) actually obtains less information than 1 bit (the true one-bit supervision should allow the querying subset to be arbitrary, not limited within one class), the effectiveness of the learning framework, as well as our multi-stage training algorithm, becomes clearer. Though we have only tested on top of the Mean-Teacher algorithm, we believe that the pipeline is generalized to other semi-supervised approaches as well as other network backbones.

Table 2 also lists three recently published methods using different backbones. We hope to deliver two messages. First, one-bit supervision is friendly to stronger backbones since it often leads to a better base model. Second, being a new learning pipeline, one-bit supervision with multi-stage training can be freely integrated into these methods towards better performance.

### 3.3    Number of Stages and Guessing Strategies

Now, we ablate the proposed algorithm by altering the guessing strategy, namely, the number of stages used, the number of fully supervised samples, the strategy of sampling $\mathcal{D}^O$, *etc*.

**First**, we compare one-stage training with two-stage training. To be specific, the former option uses up the quota of one-bit supervision all at once, and the latter uses part of them for training an intermediate model and iterates again to obtain the final model. Results on the three image classification benchmarks are shown in Table 2. The advantage of using two-stage training is clear. This mainly owes to that the intermediate model is strengthened by one-bit supervision and thus can find more positive labels than the initial model. On CIFAR100, Mini-ImageNet and ImageNet, the numbers of correct guesses using one-stage training are around 23.2K, 9.8K, and 470K, while using two-stage training, these numbers become 25.3K, 12.2K, and 475K, respectively. Consequently, from one-stage to two-stage training the accuracy of the final model is boosted by $2.63\%$, $7.24\%$, and $3.46\%$ on the three datasets, respectively.

As a follow-up study, we investigate (i) the split of quota between two stages and (ii) using more training stages. **For (i)**, Table 3 shows four options of assigning the 47K quota to two stages. One can observe the importance of making a balanced schedule, *i.e.*, the accuracy drops consistently when the first-stage uses either too many or too few quota. Intuitively, either case will push the training paradigm towards one-stage training which was verified less efficient in one-bit supervision. **For (ii)**, experiments are performed on CIFAR100. We use three training stages

Table 3: Accuracy (%) of using different partitions of quota in the two-stage training process.

| 1st-stage quota | CIFAR100 | Mini-ImageNet |
|---|---|---|
| 10K | 73.36 | 45.30 |
| 20K | 74.10 | 45.45 |
| 27K | 73.76 | 45.54 |
| 37K | 73.33 | 44.15 |

Table 2: Comparison of accuracy (%) to our baseline, Mean-Teacher [33], and some state-of-the-art semi-supervised learning methods. On all the datasets, we report the top-1 accuracy. In our multi-stage training process, we report the accuracy after initialization (using Mean-Teacher for semi-supervised learning) as well as after each one-bit supervision stage. The discussion of using one or two stages is in Section 3.3. NLS indicates negative label suppression (see Section 2.3).

| Method | CIFAR100 | Mini-ImageNet | ImageNet |
|---|---|---|---|
| Π-Model [17] | 56.57 (ConvNet-13) | - | - |
| DCT [26] | 61.23 (ConvNet-13) | - | 53.50 (ResNet-18) |
| LPDSSL [14] | 64.08 (ConvNet-13) | 42.65 (ResNet-18) | - |
| Mean Teacher [33] | 69.76 (ResNet-26) | 41.06 (ResNet-50) | 58.16 (ResNet-50) |
| Ours (1-stage base) | $51.47 \rightarrow 66.26$ | $22.36 \rightarrow 35.88$ | $47.83 \rightarrow 54.46$ |
| +NLS | $51.47 \rightarrow 71.13$ | $22.36 \rightarrow 38.30$ | $47.83 \rightarrow 58.52$ |
| Ours (2-stage base) | $51.47 \rightarrow 64.83 \rightarrow 69.39$ | $22.36 \rightarrow 33.97 \rightarrow 39.68$ | $47.83 \rightarrow 54.04 \rightarrow 55.64$ |
| +NLS | $51.47 \rightarrow 67.82 \rightarrow 73.76$ | $22.36 \rightarrow 37.92 \rightarrow 45.54$ | $47.83 \rightarrow 57.44 \rightarrow 60.40$ |

and, following the conclusions of (i), split the quota uniformly into the three stages. From the first to the last, each stage has 15K, 17K, and 15K guesses, respectively. The final test accuracy is 74.72%, comparable to 73.76% obtained by two-stage training. Though three-stage training brings a considerable accuracy gain (around 1%) over two-stage training, we point out that the gain is much smaller than 2.63% (two-stage training over one-stage training). Considering the tradeoff between accuracy and computational costs, we use two-stage training with balanced quota over two stages.

**Second**, we study the impact of different sizes of $\mathcal{D}^S$, *i.e.*, the labeled set in the beginning. On CIFAR100 and Mini-ImageNet, we adjust the size to 1K, 3K (as in main experiments), 5K, and 10K (the baseline setting with no one-bit supervision); on ImageNet, the corresponding numbers are 10K, 30K (as in main experiments), 50K, and 128K (the baseline setting), respectively. As shown in the right table, the optimal

Table 4: Accuracy (%) of using different numbers of labeled samples for the three datasets.

| # labels per class | 10 | 30 | 50 |
|---|---|---|---|
| CIFAR100 | 65.06 | 73.76 | 73.90 |
| Mini-ImageNet | 34.85 | 45.54 | 45.64 |
| ImageNet | 55.42 | 60.40 | 61.03 |

solution is to keep a proper amount (*e.g.*, 30%–50%) of full-bit supervision and exchange the remaining quota to one-bit supervision. When the portion of full-bit supervision is too small, the initial model may be too weak to extract positive labels via prediction; when the portion is too large, the advantage of one-bit supervision becomes small and the algorithm degenerates to a regular semi-supervised learning process. This conclusion partly overlaps with the previous one, showing the advantage of making a balanced schedule of using supervision, including assigning the quota between the same or different types of supervision forms.

**Third**, we investigate the strategy of selecting samples for querying. We still use the two-stage training procedure. Differently, each time when we need to sample from $\mathcal{S}^O$, we no longer assign all unlabeled samples with a uniform probability but instead measure the difficulty of each sample using the top-ranked score after the softmax computation. Here are two strategies, namely, taking out the easiest samples (with the highest scores) and the hardest samples (with the lowest scores), respectively. Note that both strategies can impact heavily on the guess accuracy, *e.g.*, on CIFAR100, the easy selection and hard selection strategies lead to a total of 30.9K and 18.0K correct guesses, significantly different from 25.3K of random sampling. Correspondingly, the final accuracy is slightly changed from 73.76% to 74.23% and 74.96%. However, on Mini-ImageNet, the same operation causes the accuracy to drop from 45.54% to 44.22% and 42.57% respectively. That being said, though the easy selection strategy produces more positive labels, most of them are easy samples and do not deliver much knowledge to the model; in comparison, the hard selection strategy obtains supervision from the challenging cases, but the number of positive labels may be largely reduced. Therefore, if the dataset is relatively easy or the model has achieved a relatively high accuracy, the hard selection strategy can benefit from hard example mining, *e.g.*, the accuracy is boosted by over 1% on CIFAR100. But, when the model is not that strong compared to the dataset, this strategy can incur performance drop, *e.g.*, on Mini-ImageNet, the accuracy drops about 3%.

In summary, in the setting of one-bit supervision, it is important to design an efficient sampling strategy so that maximal information can be extracted from the fixed quota of querying. We have presented some heuristic strategies including using multiple stages and performing uniform sampling. Though consistent accuracy gain is obtained on all three classification benchmarks, we believe that more efficient strategies exist, and may be strongly required to generalize one-bit supervision to a wider range of learning tasks and/or network backbones.

## 4 Discussions: Past and Future

### 4.1 Past: Related Work

The development of deep learning [18], in particular training deep neural networks [10, 39, 13], is built upon the need of large collections of labeled data. To mitigate this problem, researchers proposed semi-supervised learning [4, 9] and active learning [1, 20] as effective solutions to utilize the unlabeled data, potentially of a larger amount, to assist training complicated models and improve their ability of generalizing to different application scenarios.

The **semi-supervised learning** approaches can be categorized into two types. The **first** type focuses on the consistency (*e.g.*, the prediction on multiple views of the same training sample) and uses it

as an unsupervised loss term to guide model training. The $\Gamma$-variant of the Ladder Network [27] applied a denoising layer to predict the clean teacher prediction from the noisy student prediction. The $\Pi$-model [17] improved the $\Gamma$-model by removing the denoising layer and applying equal noise to the inputs of both branches. The virtual adversarial training (VAT) [22] algorithm computed a perturbation which can alter the output distribution to further improve the $\Pi$-model. The tangent-normal adversarial regularization [38] extended VAT by constructing manifold regularization. Instead of choosing the appropriate perturbation, the Mean-Teacher [33] algorithm averaged the weights of the teacher model to provide more stable targets. The **second** type assigns pseudo labels to the unlabeled samples and uses them to optimize new models. Lee *et al.* [19] took the predictions with the maximum probability as the true labels for unlabeled examples, which shared the same effect with entropy minimization [8]. Iscend *et al.* [14] employed a transductive label propagation method to generate pseudo labels on the unlabeled data for network training. To unify the advantages of two kinds of SSL methods, MixMatch [2] introduced a single loss to seamlessly reduce the entropy while maintaining consistency. *One-bit supervision is an extension to semi-supervised learning which allows to exchange quota between fully-supervised labeled and weakly-supervised samples, and we show that the latter can be more efficient.*

The **active learning** approaches [21, 37] can be roughly categorized into three types according to the criterion of selecting hard examples. The **uncertainty-based** approaches measured the quantity of uncertainty to select uncertain data points, *e.g.*, the probability of a predicted class [20] and the entropy of the class posterior probabilities [34]. The **diversity-based** approach [31, 32] selected diversified data points that represent the whole distribution of the unlabeled pool. The **expected model change** [5, 25] selected the data points that would cause the greatest change to the current model parameters, *e.g.*, BatchBALD [15] selected multiple informative points using a tractable approximation to the mutual information between a batch of points and model parameters. *One-bit supervision can be viewed as a novel type of active learning that only queries the most informative part in the class level. It is complementary to the conventional active learning and these two strategies can be integrated towards higher efficiency. Yet, as shown in Section 3.3, one needs to be careful to combine them since an overly aggressive hard example mining strategy can reduce the information obtained from the querying process.*

The idea of using human verification for the model predictions is related to the technique described in [23]. In terms of iteratively training the same model while absorbing knowledge from the previous training stage, the proposed multi-stage training algorithm is related to knowledge distillation [29, 11, 12]. It was originally designed for model compression, but recent years have witnessed its application for optimizing the same network across generations [6, 36]. *In our approach, new supervision comes in after each stage, but the efficiency of the supervision is guaranteed by the previous model, which is a generalized way of distilling knowledge from the previous model and fixing it with weak supervision.*

### 4.2 Future: Potential Research Directions

One-bit supervision is a new learning paradigm which leaves a lot of open problems.

**First**, the current one-bit supervision pipeline starts with semi-supervised learning. It would be interesting to consider starting with one-bit supervision directly (*i.e.*, $\mathcal{D}^{\mathrm{S}} = \varnothing$), which can alleviate the annotation burden but brings the challenge of considerably low guessing accuracy in the beginning. As shown in Section 3.3, this often leads to significant accuracy drop especially for challenging datasets, but its potential in further reducing the annotation workload is intriguing. We look forward to a better schedule of training stages to improve the performance.

**Second**, we have verified the effectiveness of adopting negative label suppression to extract information from the negative labels, but it remains uncovered to mining knowledge from the unlabeled data (except for using standard semi-supervision). This is particularly important when the quota of one-bit supervision is reduced and the portion of unlabeled data becomes larger. A possible way is to propagate the known labels to the unlabeled data. Note that propagating positive labels can incur considerable noise [40, 14] to harm the training procedure, but one-bit supervision has offered an opportunity for negative labels to be propagated which may have fewer errors.

**Third**, it would be interesting to extend the definition of one-bit supervision by allowing a guess to contain multiple (but still very few, in order to be practical in real-world) classes. That is to say, the question becomes *'Does the image belong to class A or class B?'* and this is considered receiving

one bit of supervision. We expect this setting can increase the flexibility of learning, but we shall emphasize that more challenges are also introduced, *e.g.* how to determine the number of classes to be queried, how to adjust the cross-entropy function to fit the 'weakly-positive' label (*i.e.*, the image belongs class A or class B), *etc*.

**Fourth**, we have investigated one-bit supervision in image classification, and the potential value of this framework lies in a broader range of vision tasks such as object detection and semantic segmentation. This is related to some prior efforts such as [24, 35]. For detection, the cost will be much lower if the labeler is given the detection results and annotates whether each bounding box is correct (*e.g.*, has an IOU no lower than a given threshold to a ground-truth object); for segmentation, similarly, the labeler is given the segmentation map and annotates determine whether each instance or a background region has a satisfying IOU. These problems are also more challenging than image classification but they have higher values in the real-world applications.

## 5    Conclusions

In this paper, we propose a new learning methodology named **one-bit supervision**. Comparing to previous approaches which need to obtain the correct label for each image, our system annotates an image by answering a yes-or-no question for the guess made by a trained model. We design a multi-stage framework to acquire more correct guesses. Meanwhile, a method of label suppression is proposed to utilize the incorrect guesses. Experiments on three popular benchmarks show that our approach outperforms the semi-supervised learning under the same bits of supervision. The idea of one-bit supervision is expected to generalized to other visual recognition tasks and non-vision learning problems. In addition, the further exploration beyond this setting, including making guesses with two or more classes at a time, remains a promising direction for the community.

**Acknowledgements**   This work was supported by the National Key Research and Development Program of China under grant 2019YFA0706200, 2018AAA0102002, and in part by the National Natural Science Foundation of China under grant 61722204, 61732007, 61932009.

## Broader Impact

This paper presents a new setting for semi-supervised learning and achieves higher efficiency of making use of annotation. We summarize the potential impact of our work in the following aspects.

- **To the research community.** The one-bit supervision setting is a new problem to the community. It raises two new challenges, namely, how to obtain more positive labels and how to learn from negative labels. We provide a simple baseline, but also notice that much room is left for improvement. We believe the study on these problems can advance the research community.

- **To training with limited labeled data.** It is an urgent requirement to extract knowledge from unlabeled or weakly-labeled data. Our work provides a new methodology that largely reduces the burden of annotation. The range of application will be even broadened after follow-up efforts generalize this framework to other vision tasks.

- **To the downstream engineers.** We provide a new framework for data annotation that can ease the downstream engineers to develop AI-based systems, especially for some scenarios in which collecting training data is difficult and/or expensive. While this may help to develop AI-based applications, there exist risks that some engineers, with relatively less knowledge in deep learning, can deliberately use the algorithm, *e.g.*, without considering the form of one-bit information, which may actually harm the performance of the designed system.

- **To the society.** There is a long-lasting debate on the impact that AI can bring to the human society. Our method has the potential to generalize the existing AI algorithms to more applications, while it also raises a serious concern of privacy, since one-bit annotation is easily collected from some 'weak' behaviors of web users, *e.g.*, if he/she views the recommended images. Therefore, in general, our work can bring both beneficial and harmful impacts and it really depends on the motivation of the users.

We also encourage the community to investigate the following problems.

1. Is it possible to generalize one-bit supervision to other forms, *e.g.*, introducing other types of light-weighted information that helps model training?

2. Are there any other solutions that can achieve higher efficiency than the proposed multi-stage training and negative label suppression methods?

3. How to generalize one-bit or few-bit supervision to other vision scenarios? In particular, what is a proper form of one-bit supervision in object detection or semantic segmentation?

## Footnotes

[2]Throughout this paper, we study the problem of image classification, yet extending one-bit supervision to other vision tasks (*e.g.*, object detection and semantic segmentation) may require non-trivial efforts.

[3]Since $\mathcal{D}^{\mathrm{O}}$ is fixed, our setting stands out from the active learning paradigm in which the algorithm mines hard examples from the unlabeled set. We will show later that one-bit supervision has different behaviors, so that aggressively mining the hardest examples does not lead to best learning efficiency.

[4]**Here is a disclaimer.** By providing 'one-bit' of supervision, the system should allow the algorithm to ask a generalized question in the form of *'Does this image belong to any class in a specific set?'* This setting is favorable to the learning algorithm especially for a weak initial model – in particular, we can start without any accurate labels. However, asking such questions can largely increase the workload of labelers, so we have constrained the query to be a single class rather than an arbitrary set of classes. From another perspective, it takes a labeler much more efforts to provide the accurate label of an image (more than $\log_2 C \times$ that of making a one-bit annotation). Therefore, the 'actual' ratio of supervision between full-supervision and one-bit supervision is larger than $\log_2 C : 1$ – in other words, under the same supervision bits, our approach actually receives a smaller amount of information.

[5]To guarantee the correctness of normalization (*i.e.*, the scores of all classes sum to 1), in practice, we set the *logit* (before softmax) of the negative class to a large negative value.

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
