[Reviews · NeurIPS 2020]

Review 1

Summary and Contributions: This paper proposes an iterative weakly-supervised active learning method that learns from the one-bit label annotating whether the model prediction is correct or not. In each iteration, it selects part of the unlabeled set for querying and updates the model with the queried annotation. Experiments show that the proposed model performs better than the semi-supervised baseline when both are given the same bits of labeling information.

Strengths: 1. The paper is well-written and easy to read. The literature review is comprehensive and the motivation and method are clearly stated. 2. The one-bit supervision setting is novel from my knowledge. 3. The ablation studies are helpful for readers to understand how the method works. The observations about the optimal partition between two stages and between fully supervised and one-bit supervised are both interesting.

Weaknesses: 1. (Main concern) The paper compares the proposed one-bit supervision with full supervision given the same amount of information bits. However, the number of information bits is not necessarily proportional to the labeling time. For example, in practice labeling 3 images with one-bit labeling could take more time than labeling one image with 8 classes. The paper would be more convincing if the labeling time of different types of supervision can be shown like in [1]. 2. The proposed model still needs some fully-labeled samples to train the initial model. It would be interesting to show how the model would perform with only one-bit labels. Minor: L18: belong -> belongs L27, 28: gives -> give L341: in a time -> at a time [1] Bearman, Amy, et al. "What’s the point: Semantic segmentation with point supervision." European conference on computer vision. Springer, Cham, 2016. -----------Update------------ The additional info authors provide has addressed my main concern. I'm raising my score.

Correctness: yes.

Clarity: yes.

Relation to Prior Work: yes.

Reproducibility: Yes

Additional Feedback:


Review 2

Summary and Contributions: This paper proposes a novel weakly-supervised learning methodology, named one-bit supervision, for image classification which just needs the annotator to verify whether an image belongs to a specified class. In order to acquire more correct guesses, it starts with a semi-supervised method and makes use of a multi-stage training paradigm to use the supervision quota. Meanwhile, a method of label suppression is proposed to utilize incorrect guesses. Experiments on three popular benchmarks show that the proposed method outperforms the semi-supervised learning under the same bits of supervision.

Strengths: 1. The idea that annotating an image by answering a yes-or-no question is very interesting. 2. This new learning methodology seems easy to apply in the real world. 3. Sufficient experiments on three image classification benchmarks verify the effectiveness of the proposed method. 4. The proposed multi-stage training framework and the method of negative label suppression effectively improve the performance of the proposed method, which is verified in the ablation study. 5. The paper is well written and technically are clearly explained.

Weaknesses: 1. The idea that doing extra annotations using the model predictions is related to the technique described in “We don’t need no bounding-boxes: Training object class detectors using only human verification”, which is ignored by the authors. 2. How do you handle the samples selected twice in the second training stage? Please explain in detail. 3. The results on the Imagenet dataset have not achieved the SOTA performance. 4. There are no experiments in the paper to accurately verify whether the proposed method can reduce the annotation cost.

Correctness: Yes, the claims and methods are correct.

Clarity: Yes, the paper is well written.

Relation to Prior Work: Yes, compared to previous approaches that need to obtain the correct label for each image, the proposed system annotates an image by answering a yes-or-no question for the guess made by a trained model.

Reproducibility: Yes

Additional Feedback:


Review 3

Summary and Contributions: In the context of semi-supervised learning and active learning, the paper proposes the one-bit supervision learning strategy for image classification. The idea is to leave a portion (in the extreme case could be all or none) of the labeled data in a semi-supervised learning setting to be guessed with yes/no answer for a class for each sample, instead of directly learning with the ground truth labels (i.e., the full bit learning). Two issues are studied: how to make a good guess and how to make use of the negative samples (those guessed wrongly). Solutions of a multi-stage training with partitioning the portion into multiple parts for iteratively improving guessing accuracy and a negative label suppression to artificially force the score of the negative samples to zero. Evaluations are reported from three benchmarks to demonstrate the effectiveness of this approach.

Strengths: 1. The work is certainly relevant to the NeurIPS community 2. The proposed method appears to be technically correct and advances the SOTA 3. The evaluations appear to be extensive and convincing 4. The paper is well-written and the presentation is clear and easy to follow, except for a few typos and grammatical errors 5. The proposed method is simple and easy to implement 6. Though not directly stated in the paper, the paper is inspiring to a broader research question. See my comments below.

Weaknesses: 1. The novelty of the proposed method is arguably limited. See my comments below. 2. There are minor issues in presentation

Correctness: Yes

Clarity: Yes

Relation to Prior Work: Yes

Reproducibility: Yes

Additional Feedback: I consider this work as a new method in the context of semi-supervised learning and actively learning. Indeed, these are the two topics the authors of the paper reviewed as the related work to this work. The method essentially is yet another way to rearrange labeled samples and unlabeled samples in order to identify “active” samples to improve the learning accuracy. Thus, it is not an eye-opening, truly novel approach. I would argue that this method is incrementally novel at best. On the other hand, the method is pretty simple and easy to implement, and also it appears to be working, which is considered as an advantage. This work, however, brings up a broader, more interesting question that is worth investigating. That is, given a semi-supervised setting of a dataset and a semi-supervised learning method, is there anyway to further raise up the learning efficacy through the manipulation and rearrangement of the samples in the dataset (but without changing the original setting) by introducing a new mechanism (such as the one-bit supervision mechanism introduced in this paper)? The presentation in general is good and easy to follow. There are a few typos and grammatical errors. Table 4 appears to be incomplete per the discussions in the text. The above was my original review. After the authors uploaded their rebuttal, I read it carefully. The authors addressed all of my concerns I stated in my original review, though I was still not convinced that this work was a piece of truly innovative contribution. So I stay with my original review and overall rating. I also looked at the other reviews and I think all the reviews are more or less converged. By the way, in the authors' rebuttal, it appears that they did not use cut and paste from my original review. They had a typo in copying my original statement: more interesting, not mode interesting.


Review 4

Summary and Contributions: After carefully reading the authors' rebuttal and the other reviews, my opinion of the paper is not changed. The authors addressed my concerns but I still not fully convinced that this is truly innovative contribution. Nonetheless, I believe it is worth of discussion for the community. Hence, my overall rating is unchanged. ---- The paper is about a novel way to exploit supervision in an active learning setting when training classifiers, i.e with a given fully annotated training set and another set of images that can be ask to be partially labeled. Given that annotating datasets is costly, the idea is to reduce the burden on annotators by shifting from the question "which class this image belongs to?" to "does this image belong to this specific class x?". Asking if an image belongs to a class is considered less costly for an annotator but reduce the amount of bits of supervision available for training a classifier. The classifier is updated multiple times during the whole process of training the system. The paper contains the proposal of this semi-supervised problem, considerations about when and how many times a classifier should be updated during the process, how to exploit the one bit supervision in case of negative guess. Experiments show the importance of using both the positive and negative guesses in the known classifier Mean-Teacher, able to reach state of the art performance on three public datasets. Ablations studies clarifies basic choices of how many stages, which images should be guessed between random or easy/hard images, the size of the starting fully annotated dataset. Discussion on past and future work, provides an outline of possibile directions from this contribution.

Strengths: + The paper is easy to read and well presented. I appreciated the discussion of past and future work in relation to the contribution. It is definitely of relevance for the neurips community. + The contribution is novel for the active learning approaches, while also related to semi-supervised learning. + The idea is simple and execution straight forward, which is a plus.

Weaknesses: - The fact that it is simpler and cheaper to obtain one bit supervision versus full supervision is not explored and given as hypothesis from the introduction section. However, intuitively it may be that obtaining one bit supervision can be as expensive as full supervision. For instance, I think about the expertise of an annotator: if he/she immediately recognize the class of an image, it is straightforward to say yes or no, but it is also straightforward to say which class it is, so they have the same cost. Besides [29][40] which shows that it is hard for a human to memorize all classes in the case of ImageNet, it would be important to show which are the scenarios where one bit supervision is actually useful. A realistic study of annotating a dataset with this technique would be interesting to show. - The paper is posed more as a well executed proof of concept with limited scope which is left for the community. Experiments tests the Mean-Teacher method only, which can be deemed enough for showing potential of the idea.

Correctness: The method appear to be generally correct and methodologically correct. - In sec 3.3, while intuitively the first stage quota seems the most important (for (i)), the results in table 3 show that the accuracy is comparable in all cases (for instance on CIFAR100, 73.36 % vs 74.10 vs 73.76% and 73.33%), as similarly stated for (ii) when 73,76% is comparable to 74,72% when changing the number of stages. Hence, I do not agree that the more stages experiments show comparable results while table 3 show not comparable results...

Clarity: The paper is easy to read and well presented.

Relation to Prior Work: The paper contain a throughly discussion of past and future work.

Reproducibility: Yes

Additional Feedback: - For the sake of precision, in Figure 1 the blue triangles are concentred in a separate area with respect of the dark circles. It may give the impression that the training set is completely separate from the other sets, while in reality they have similar distribution. Hence they should be shuffled together. - Table 4, the labels per class should be 10% 30% 50% ?

[Author Response · NeurIPS 2020]

We thank all reviewers for their valuable comments. This paper presents a novel weakly-supervised learning approach in which the model keeps predicting the class of each unlabeled sample and learns from the feedback that whether the prediction is correct. The reviewers are generally satisfied with our writing and experiments. The most important concern (**R1**, **R2**, **R4**) lies in whether the annotation cost is indeed related to the number of supervision bits – we answer it in the common part. Other concerns are minor – we carefully respond below and will revise the paper accordingly.

**Q0:** *Can one-bit supervision reduce annotation costs? Any study on the actual comparison between full supervision and one-bit supervision?* **A0:** Thanks for this question! We asked three labelers to annotate 100 images from ImageNet using the one-bit setting, and the average annotation time is 2.72 seconds per image (with a precision of 92.3%). According to the authors of ILSVRC2012 [Russakovsky et al., IJCV'15], the average time for a full-bit annotation is around 1 minute, much higher than $10\times$ of the one-bit cost. This validates our motivation in a many-class dataset.

**Response to Reviewer #1**

**Q1:** *# of information bits not necessarily proportional to labeling time?* **A1:** Please refer to the common question.

**Q2:** *How the model would perform with only one-bit labels?* **A2:** Very good question! Using pure one-bit supervision will lead to lower accuracy than the mixed schedule (first using fully-supervised samples and then using one-bit-supervised samples) used in the paper. This aligns with our diagnosis in Section 3.3 showing that the best performance is achieved under a balanced configuration (a moderate number of fully-supervised samples). We agree that pure one-bit supervision is interesting in research (Section 4.2), we point out that in practice, it is reasonable to label a small number of fully-supervised or make use of existing annotation, so that the scenario is closer to that studied in the paper.

**Q3:** *Some minor grammatical mistakes?* **A3:** Thanks for the kind reminder! We will fix them in the final paper.

**Response to Reviewer #2**

**Q1:** *Omitted related work?* **A1:** Thanks for the kind reminder. We will cite and discuss this paper in Section 4.1.

**Q2:** *How to handle the samples selected twice in the second training stage?* **A2:** Thanks for the question! For each sample in Stage 2, if it is a wrongly predicted sampled in Stage 1, we first guarantee that the previously guessed label is not guessed again. Then, if the guess in Stage 2 is correct, this sample gets the true label, it is removed from $\mathcal{D}^{O-}$ and added to $\mathcal{D}^{O+}$. If the guess wrong again, it has two negative labels and still stay in $\mathcal{D}^{O-}$.

**Q3:** *Results on ImageNet have not achieved SOTA?* **A3:** We agree. We have used Mean-Teacher as the baseline model and demonstrated improvement over it. We believe our pipeline can benefit other semi-supervised models and even self-supervised-then-semi-supervised models. We will try to report more results in the final paper.

**Q4:** *Whether the proposed method can reduce the annotation cost?* **A4:** Please refer to the common question.

**Response to Reviewer #3**

**Q1:** *This work is incrementally novel, however, it brings up a broader, mode interesting question...* **A1:** Thanks for the comments! Our work can be considered as reducing the basic unit of supervision, that is, from an entire sample that can contain multiple bits to a single bit. As mentioned by the reviewer, our method is easy to implement and works well, which inspires the community to study a new mechanism of improving the efficacy of learning and annotation. We believe this idea as well as our simple baseline is worth announcing to the community to facilitate future research.

**Q2:** *A few typos and grammatical errors?* **A2:** Thanks for the kind reminder! We will fix them in the final paper.

**Q3:** *Table 4 appears to be incomplete?* **A3:** Sorry for misleading. The 'missing' contents of Table 4 are the baseline results which we have provided in Table 2. We will add them back to Table 4 to avoid misleading. Thanks!

**Response to Reviewer #4**

**Q1:** *A realistic study of annotation with this technique would be interesting.* **A1:** Please refer to the common question.

**Q2:** *The paper is a well executed proof of concept with limited scope.* **A2:** We agree with your summary. The current status of this work is a proof-of-concept in image classification, and we believe that the idea as well as our simple baseline can inspire the community for future work (see Section 4.2). While the reviewer accepted that 'our experiments can be deemed enough for showing potential of the idea', we will try to generalize our algorithm to more models/tasks.

**Q3:** *Confusion of the statements in Section 3.3.* **A3:** Sorry for misleading. (i) We hope to highlight the impact brought by the 1st-stage quota, which is shown as a $0.7\%$–$0.8\%$ gap in accuracy, and we consider it significant. (ii) We will rewrite the statement as follows. Though 3-stage training brings a considerable accuracy gain (around $1\%$) over 2-stage training, we point out that the gain is much smaller than $2.63\%$ (2-stage training over 1-stage training). Considering the tradeoff between accuracy and computational costs, we use two-stage training with balanced quota over two stages.

**Q4:** *Issues in Figure 1 and Table 4?* **A4:** Thanks for the reminder! For Figure 1, we will use a better representation to avoid misunderstanding. For Table 4, they are indeed the number of training samples per class – please refer to Table 1.

[Meta-Review · NeurIPS 2020]

The paper proposes a new paradigm for image annotation called one-bit supervision based on questioning whether a random image belongs to a predicted category or not. Under the assumption that annotating an image with K categories is as expensive as log K annotations of the form of one-bit supervision, the paper shows that multi-stage semi-supervised learning using one-bit supervision is more effective than standard semi-supervised learning under the same annotation costs. The setup is interesting and convincing as the first step, but as the reviewers noted, the clarity of exposition and claims can improve. Also, it is worth elaborating whether you use softmax cross-entropy loss as mentioned in L112 or L2 loss in Eq. (1).